# Activity of Carbonic Anhydrase VI is Higher in Dental Biofilm of Children with Caries

**DOI:** 10.3390/ijms20112673

**Published:** 2019-05-30

**Authors:** Daniele C. R. Picco, Lenita Marangoni-Lopes, Thaís M. Parisotto, Renata Mattos-Graner, Marinês Nobre-dos-Santos

**Affiliations:** 1Piracicaba Dental School, Pediatric Dentistry Department, University of Campinas, 901 Limeira Av, Piracicaba 13414-903, São Paulo, Brazil; danielepicco@outlook.com (D.C.R.P.); lenita_m_l@hotmail.com (L.M.-L.); 2Laboratory of Microbiology and Molecular Biology, University São Francisco, 218 São Francisco de Assis Av, Bragança Paulista 12916-900, São Paulo, Brazil; thaisparisotto@gmail.com; 3Piracicaba Dental School, Microbiology Department, University of Campinas, 901 Limeira Av, Piracicaba 13414-903, São Paulo, Brazil; rmgraner@fop.unicamp.br

**Keywords:** carbonic anhydrase VI, biofilms, dental care for children

## Abstract

This study investigated pH, activity and concentration of carbonic anhydrase VI (CA VI) in dental biofilm of caries and caries-free children of 7–9 years old. Seventy-four children were selected and divided into two groups. The caries diagnosis was performed according to the WHO criteria, including the early caries lesion. After biofilm collection and pH determination, CA VI concentration and activity were determined by ELISA and Zimography respectively. The data were submitted to a Mann-Whitney test and to Pearson and Spearman correlation analyses. Means and standard deviations of dental caries for the caries group were of 3.162 ± 1.385. The biofilm pH was significantly higher in the caries-free group. The CA VI activity was significantly higher in biofilm of children with caries. The CA VI concentration was significantly higher in biofilm of caries-free children. In caries-free children, there was a moderate negative correlation between CA VI activity and concentration in dental biofilm as well as between pH and CA VI activity. A negative correlation between biofilm pH and CA VI concentration was found in the caries group. In conclusion, CA VI was shown to be more active in the biofilm of school children with caries in order to contribute to neutralization of biofilm acid.

## 1. Introduction

According to scientific evidence, dental caries is a sucrose-biofilm-dependent and multifactorial disease. Among the factors that modulate this disease, the *Streptococcus mutans* plays an important role on the onset of dental caries. In addition, the frequency of sugar exposure as well as the chemical composition of the teeth and the salivary properties like buffer capacity and flow rate are also important factors [1,2]. Imbalance or dysbiosis within the plaque biofilms is the starting event that leads to major oral diseases, such as caries and periodontal disease [3]. The presence of a thick dental biofilm on the maxillary incisors has been reported as a caries risk indicator because in this undisturbed biofilm, the ions exchange between saliva and biofilm is strongly impaired [4]. Given the importance of salivary properties to oral health, it is well known that rampant caries is observed in individuals with decreased salivary function. Moreover, the maintenance of oral homeostasis is provided by the salivary buffering capacity. This property of saliva is mainly accomplished by the phosphate and bicarbonate systems. However, in stimulated saliva, the total buffer capacity of saliva is granted by the bicarbonate buffer. Thus, during the caries dynamic process, the bicarbonate ions can neutralize lactic and acetic acids produced by cariogenic bacteria of biofilm and hamper demineralization [5]. In this regard, previous investigations showed that carbonic anhydrase VI (CA VI) appears to be one of the key enzymes in the oral physiology of humans and animals [6,7,8]. Once secreted into saliva, CA VI is able to grant enhanced buffering capacity and to support a reservoir of HCO_3−_ in the salivary glands [9]. 

Concerning the action mechanism of CA VI in saliva, previous research carried out by Ozturk et al. [10] compared caries and caries-free young adults and detected no significant difference in the CA VI concentration in their saliva. However, a low CA VI concentration was demonstrated to be associated with a higher caries index according to Kivela et al. [11]. In the same way, Szabó [12] demonstrated that saliva of 7 to 14 year old caries-free children contained a higher concentration of CA VI than saliva of children with caries. On the other hand, the early work from Frasseto et al. [13] used the zymography analysis [14] and quantified the activity of salivary CA VI. The results of this study showed a significantly higher CA VI activity in pre-school children with early childhood caries than in caries-free children after a 20% sucrose rinse. 

In addition, further evidence on the relationship between caries and CA VI activity in saliva was recently provided by the longitudinal study of Borghi et al. [15].

Regarding dental biofilm, the early investigation performed by Kimoto et al. [7] employed the Western blot analysis in order to demonstrate that CA VI was concentrated in dental biofilm. Regarding further evidence of the CA VI effect in buffering biofilm pH, these authors demonstrated that when a CA-specific inhibitor (acetazolamide) was added to a sucrose solution, a significant decrease in biofilm pH occurred. However, being concentrated in dental biofilm may not necessarily mean that all of the CA VI isoenzyme present in the media is active. Furthermore, knowledge regarding the activity of CA VI in dental biofilm, as well as its relationship with dental caries, is unknown. Thus, determining the activity of CA VI instead of just its concentration would provide further evidence of the effect of this isoenzyme in buffering biofilm pH, which consequently would protect teeth from dental caries. Moreover, another relevant aspect that deserves further investigation is whether there is any relationship between the activity and concentration of CA VI in the biofilm of children with caries, as well as between biofilm pH and activity and concentration of CA VI.

Considering the relevance of dental biofilm in the dynamic of the caries process, the aim of this study was to investigate whether CA VI activity was higher in biofilm of 7–9 year-old children. In addition, we also investigated the relationship among dental caries, activity and concentration of CA VI, and biofilm pH in 7–9 year-old school children.

## 2. Results

### 2.1. The DMFT Index

The mean and standard deviation of the number of decayed, missing and filled teeth for the group with caries were 3.162 ± 1.385. According to the World Health Organization, this caries index is considered moderate. The caries-free group had no caries lesion.

### 2.2. Dental Biofilm pH

Figure 1 shows that biofilm pH was significantly higher in the caries-free group (6.155 ± 0.4839 *n* = 37) than in the group of children with caries (5.666 ± 0.5534 *n* = 37; *p* = 0.0005). The values refer to medians and interquartile deviations.

### 2.3. CA VI Activity and Concentration in Biofilm

Figure 2 shows that the CA VI activity was significantly higher in biofilm of children with caries (25.96 ± 16.41 *n* = 34) than in the caries-free children (17.65 ± 9.52 *n* = 33; *p* = 0.0421). This Figure also evidenced that CA VI concentration was significantly higher in the biofilm of caries-free children (3.507 ± 4.014 *n* = 25) than in the caries group (1.693 ± 1.802 *n* = 26; *p* = 0.0335). The values refer to medians and interquartile deviations.

### 2.4. Correlation among the Independent Variables in Caries and Caries-Free Children

The results of Table 1 show that in caries-free children there is a moderate negative correlation between CA VI activity and concentration in dental biofilm (*p* = 0.02068) as well as between pH and CA VI activity (*p* = 0.00683). Regarding the caries group, we detected a negative correlation between biofilm pH and CA VI concentration (*p* = 0.0164). 

## 3. Discussion

As a dynamic disease, dental caries results from the imbalance between demineralization and remineralization of enamel that occurs as a consequence of frequent sugar exposure that enables the production of organic acids by oral microorganisms [16]. After exposure to sucrose and to a lesser extent starch, salivary constituents can control biofilm pH, thus enhancing the caries prevention effect. In addition, saliva washes out dietary sugars and acids derived from microorganisms metabolism but its role in buffering pH in the oral cavity is mainly due to the carbonate and phosphate buffers that neutralize acid [7].

After sucrose intake, glycolysis is initiated, and acids are produced by bacteria in the biofilm. In the biofilm-enamel interface, an increase in the hydrogen ions concentration occurs, then local pH starts to fall [17]. This pH drop should be stopped by the action of biofilm buffer fluid elements such as saliva to account for the buffering of plaque acid. Considering that in stimulated saliva the main buffering effect is performed by bicarbonate, one should expect a higher outcome of saliva in controlling the pH drop of biofilm. 

Thus, investigations into the concentration and activity of CA VI in dental biofilm is relevant to clarify how this isoenzyme behaves in children with caries. 

To figure out how CA VI contributes to the protection of enamel surface from dental caries, we investigated for the first time the activity of this isoenzyme in dental biofilm. Dental plaque is the biofilm naturally found on the teeth surface and it is also implicated in dental caries. 

This disease results from the interaction of specific bacteria like *Streptococcus mutans* and other microorganisms with dietary components. Increased proportions of *Streptococcus mutans* causes a microbial imbalance of the biofilm due to the acidogenic and acid-tolerant characteristics of this microorganism [18]. 

The results of this study demonstrated that CA VI is not only concentrated in biofilm, but more importantly, the isoenzyme is active in order to increase acid neutralization after the frequent cariogenic challenges that children with caries are expected to be exposed to. Indeed, CA VI activity was significantly higher in the biofilm of children with caries than in the biofilm of caries-free children. This result suggests that the isoenzyme is being more constantly activated in individuals with caries as a protective mechanism to neutralize the pH of the medium, specifically of dental biofilm, to provide a greater defense against enamel demineralization. In line with this assumption, the early investigation performed by Frasseto et al. [13] showed that before a 20% sucrose rinse, CA VI activity and its variations were higher in saliva of children with caries than in saliva of caries-free children. These findings were confirmed in a recent longitudinal investigation carried out by Borghi et al. [15]. In addition, our research group demonstrated a 4.88 times higher likelihood for caries development when saliva of this children exhibited a CA VI activity higher than 1.75 [19].

The findings of the current investigation suggest that CA VI can provide protection against dental caries by activating the bicarbonate buffer in the oral medium. In this way, CA VI isoenzyme enhances the acid buffering at the biofilm–enamel interface [11]. In saliva, bicarbonate diffuses into enamel pellicle and into dental biofilm, and binds to hydrogen ion to immediately produce carbonic acid. Consequently, this H_2_CO_3_ dries up to carbon dioxide and water; however, the ordinary degree of these chemical events is considered to be not fast [20].

However, CA VI is known to increase the speed of the dehydration reaction by up to 13,000 times [21]. Our results give support to this mechanism in that we found a negative correlation between salivary CA VI activity in saliva and in dental biofilm (r = −0.3910; *p* = 0.0360 data not shown). Therefore, it is proposed that CA VI in dental biofilm contributes to the neutralization of biofilm acid in the microenvironment of the tooth surface as a consequence of the buffering performed by salivary bicarbonate and, thus, may help to prevent dental caries [7,20]. Nevertheless, different from our recent work demonstrating the power of CA VI activity in saliva to predict caries development [19], we were not able to show the same ability of CA VI activity in biofilm. A possible explanation for this finding could be that data of CA VI in biofilm showed a higher variation as compared with CA VI activity in saliva even though an appropriated number of individuals was used as evidenced by the sample power calculation which showed a power of 0.80.

An interesting finding of the current investigation was that we observed a negative correlation between CA VI activity and concentration in biofilm, happening only in caries-free children. Furthermore, in this group of children we also found a significant negative correlation between biofilm pH and CA VI activity (*p* = 0.0068), which means that the higher the biofilm pH, the lower the activity of the isoenzyme. These results can be explained when considering that caries-free individuals are less frequently exposed to cariogenic carbohydrates and, consequently, to lower pH falls in dental biofilm [22,23], which in turn would activate the CA VI concentrated in dental biofilm [24]. Our data give support to this inference, since we showed that in caries-free children, biofilm pH was significantly higher than in children with caries (*p* = 0.0005). In this way, the acid buffering in dental biofilm provided by the activity of CA VI would not be so necessary in caries-free children. 

Regarding the caries group, our results demonstrated a negative correlation between biofilm pH and CA VI concentration, even with a higher concentration being observed in the caries-free group. This finding suggests that, in caries-active children, when proton concentration increases, as a consequence of the biofilm pH drop, the expression of this isoenzyme becomes greater in order to regulate the actual biofilm pH. This pH regulation would be a consequence of bicarbonate increase in dental biofilm since CA VI is known to catalyze the reversal reaction CO_2_ +H_2_O ↔ H^+^ + HCO_−3_ [24].

As a limitation, in this study, biofilm samples were collected from all accessible vestibular surfaces of incisors and molars, be they healthy or carious. Note that since children were between 7 and 9 years of age, they are not expected to have premolars. In relation to biofilm examination of early caries lesion and cavitated carious lesion, there is a possibility that CA VI activity may change/increase according to caries severity, and this information would provide an additional view on how this isoenzyme behaves. Besides that, we did not investigate the concentration and activity of CA VI in biofilm at follow-up to better understand the behavior of this isoenzyme as dental caries remineralizes or progresses. Thus, longitudinal studies investigating the CA VI activity at baseline as well as at follow-up are strongly encouraged and are ongoing in our laboratory. In addition, in the present study, the biofilm pH was determined out of the oral cavity and it is known that the buffering of saliva plays an important role in the biofilm pH [25].

Another result of this study was that biofilm pH was significantly higher in caries-free children than in children with caries. Our results are in accordance with Appelgren et al. [26], who claimed that caries-free individuals had a higher baseline biofilm pH in the anterior mandibular region, compared to caries-active individuals. Although there was no information on the frequency of acidogenic episodes that children of this study were exposed to, this finding would be expected if we consider that caries-free children are less exposed to rapidly fermentable carbohydrate, usually sucrose with a lower frequency than children with caries [22]. In line with this assumption, the early work of Armfield et al. [27] demonstrated a significant association between caries and sugar-sweetened beverage consumption. On the other hand, our data are not in line with previous investigations from Peres et al. [5], Kimoto et al. [7] and Frasseto et al. [13] who found no change in biofilm pH in children with caries and caries-free children. However, it should be noted that these authors determined the biofilm acidogenicity after a sucrose rinse, an experimental condition distinct from the one we used in the present investigation.

Caries-free individuals or those with a minimum of cavities tend to show a biofilm pH at a slightly higher break, a higher pH per min after consumption of fermentable carbohydrates, and faster return to resting levels when compared to individuals susceptible to cavities. When saliva is depleted, however, the differences between caries-free individuals and individuals more susceptible to dental caries are less marked, and the minimum values achieved by pH are lower. These findings indicate the importance of saliva as a determinant of susceptibility to cavities by modifying the biofilm pH response [28].

It can be concluded that carbonic anhydrase VI was shown to be more active in dental biofilm of school children with caries in order to contribute to the neutralization of biofilm acid. The higher pH and concentration of CA VI in dental biofilm in caries-free children suggests that, since these individuals are not so frequently exposed to cariogenic challenges, their biofilm pH drop does not constantly occur to activate the bicarbonate buffer.

## 4. Materials and Methods

### 4.1. Ethical Considerations

This study was approved by the Research Ethics Committee of Piracicaba Dental School/University of Campinas (protocol number 018/2012, approval date: 17/09/2012). All procedures performed in studies involving human participants were in accordance with the ethical standards of the institutional and/or national research committee and with the 1964 Helsinki declaration and its later amendments or comparable ethical standards. Parents or guardians who agreed with the inclusion of the children in the study completed and signed a free and informed consent form, and the children selected also agreed to participate in the study.

### 4.2. Sample

The sample power calculation (95%; α = 5%) was based on the study previously performed by our group [13], which used similar methodology, and found that the activity of CA VI in the saliva of caries and caries-free pre-school children was of 42.75 ± 32.47 and 19.13 ± 16.391, respectively. Sample size was based on the averages for two independent samples and bilateral test (parametric test) by the equation: $n=\frac{\left({s_{1}}^{2}\;+\;{s_{2}}^{2}\right){\left(z_{1}\;-\;\frac{\alpha }{2}\;+\;z_{1}\;-\;\beta \right)}^{2}}{{\left({\bar{x}}_{1}\;-\;{\bar{x}}_{2}\right)}^{2}}$ = 30.78. In this way, 31 subjects were allocated to each group. As a cross-sectional investigation was conducted, the estimated number of subjects (31) was increased by 20% to compensate for possible subject drop-out rate. To make sure that children with caries as well as those who were caries-free had the same chance to be selected for the study, they were randomly selected, one by one, using the Excell lottery method, and examined for caries diagnosis. After finding 37 caries-free children, the process was repeated to select 37 children with caries. Children were considered to be caries-free if their number of decayed, missing or filled teeth (DMFT) was equal to zero. On the other side, children who had a DMFT ≥ 1 were allocated in the caries group. Thus, we selected 74 school children aging 7 to 9 years old from a public school from the city of Piracicaba, SP, Brazil. The selected children were healthy, had good oral hygiene, good record of general and oral health, and were not taking any medication. All children had healthy gums with no redness, swelling or bleeding.

### 4.3. Calibration of the Examiner and Caries Diagnosis

Before clinical examinations, theoretical discussions using clinical photographic slides were held to give instructions to the examiner about the use of criteria and the examination method. Following, practicing exercises were carried out and replicate examinations were performed on a random sample of 14 schoolers, with a 1-week interval period. This step of training exercises lasted 30 h. The intra-examiner agreement regarding all the DMFT components, including the early caries lesion (ECL) was measured using Kappa calculation. Agreement at the tooth level was of 0.68.

Caries diagnosis was carried out using the visual inspection method with the aid of a clinical mirror, a CPITN probe and flashlight after cleaning and drying the teeth with gauze, according to the World Health Organization criteria and adding the early caries lesion [29].

### 4.4. Biofilm pH Determination

Children were instructed to interrupt their tooth brushing procedures 48 h before collections. Biofilm samples (about 1 mg) were obtained with a sterile inoculation loop from all accessible buccal surfaces of incisors and molars. Each biofilm sample was suspended in 20 µL of distilled water in a plastic micro eppendorf and frozen at −40 °C until subsequent analyses. The biofilm pH was read with a glass combination electrode (PerpHect^®^ ROSS^®^; Thermo Scientific Orion, Waltham, MA, USA), previously calibrated with standard buffer solutions at pH 4 and 7. 

### 4.5. Determination of CA VI Activity in Dental Biofilm

The determination of CA VI activity was performed with the use of the modified protocol of Kotwica et al. [14], adapted to CA VI in dental biofilm in our laboratory. For this analysis, 2 µg of dental biofilm was collected and suspended in 40 µL of the electrophoresis sample buffer and frozen at −40 °C. After being thawed, the biofilm sample (50 µL) together with the electrophoresis sample buffer (10 µL) were stirred at 12%/0.8% bisacrylamide, and ran for 2:30 h, at 140 V and at 4 °C. Following, each channel was filled with this material. After electrophoresis, the gel was incubated for 10 min, in 0.1% bromothymol blue diluted in 100 mmol/L Tris buffer, pH 8.2. For CA VI activity determination, gels were immersed in distilled deionized water saturated with CO_2_. Subsequently, photographs were obtained from the gels, and the images were evaluated using the ImageJ^®^ software [30] to calculate the radiance of the band area. Results of CA VI activity are shown as pixels area.

### 4.6. Determination of CA VI Concentration in Dental Biofilm

For this analysis, initially, protein extraction from the biofilm was performed. To each sample (2 µg of biofilm), we added 200 mg of silica beads and 130 µL of distilled water. This sample was stirred in a bead better apparatus for 1 and a half minutes (45 s ON—1 min ice—45 s ON—1 min ice), and the volume having no silica beads was then transferred to an eppendorf tube and stored at −40 °C until subsequent analysis. Analysis of the CA VI concentration in dental biofilm was performed using an Enzyme-linked Immunosorbent Assay Kit for carbonic anhydrase VI (Elisa kit SED073Hu 96 tests—Cloud-Clone Corp., Katy, TX, USA) following the manufacturer’s instructions. After the experimental protocol of the kit, the absorbance was evaluated using an EON spectrometer (Biotek Instruments, Winooski, VT, USA) and the concentration of CA VI was expressed as a nanogram of CA VI per microliter of biofilm (ng/µL).

### 4.7. Statistical Analysis

To perform the statistical analysis, data normality was checked using the D’Agostino-Pearson test. For this analysis, the dependent variables were caries-free children and children with caries. The activity and concentration of CA VI as well as the biofilm pH were considered independent variables. Data of pH, activity and concentration of CA VI of dental biofilm did not follow normal distribution and were analyzed using the Mann-Whitney test. In addition, the correlation between dental caries and all independent variables under study as well as the correlation between CA VI activity and CA VI concentration, biofilm pH and CA VI activity, and biofilm pH and CA VI concentration were assessed using the Pearson and Spearman correlation analyses. The statistical tests would be significant if *p*-values were less than 0.05. Data were analyzed using the GraphPad Prism 6.01 (GraphPad Software Inc., San Diego, CA, USA). 

## Figures and Tables

**Figure 1 ijms-20-02673-f001:**
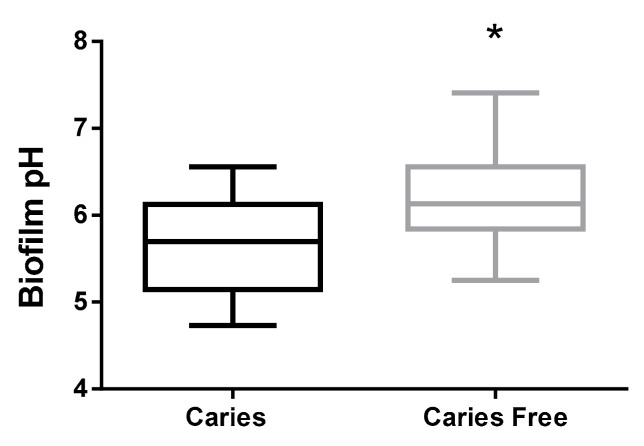
Boxplot graphics of biofilm pH, expressed as pH units of caries and caries-free children. (*p* = 0.0005, Mann Whitney test). ***** Means statistically significant difference between groups.

**Figure 2 ijms-20-02673-f002:**
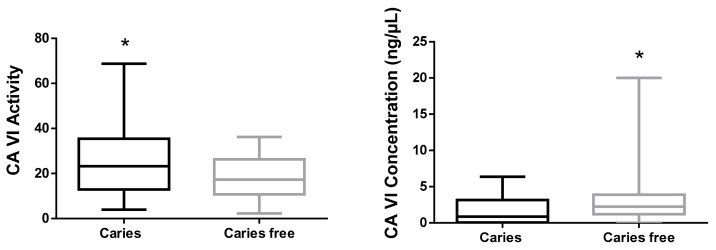
Boxplot graphics of CA VI activity (CA VI) (*p* = 0.0421, Mann Whitney test) and of CA VI concentration (*p* = 0.0335, Mann-Whitney test) in dental biofilm of caries and caries-free children. ***** Means statistically significant difference between groups.

**Table 1 ijms-20-02673-t001:** Correlation between carbonic anhydrase (CA VI) activity and concentration, biofilm pH and CA VI activity, and biofilm pH and CA VI concentration in biofilm of children with caries and caries-free children. Values refer to Pearson and Spearman correlation coefficients.

Variables	Groups of Children			
	Caries		Caries-Free	
	r	*p*	r	*p*
CA VI Activity × Concentration	0.04288	0.8352 ^S^	−0.460	0.02068 ^S^
pH × CA VI Activity	−0.0113	0.9477 ^S^	−0.4551	0.00683 ^P^
pH × CA VI Concentration	−0.4657	0.0164 ^S^	0.2196	0.29132 ^S^

P Pearson correlation for normal data. S Spearman correlation when data did not follow normal distribution. r = correlation coefficients and *p* = probabilities of statistical significance.

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
