# Peer review of "Activity of Carbonic Anhydrase VI is Higher in Dental Biofilm of Children with Caries"

_ijms, 2019, doi:10.3390/ijms20112673_

Round 1

Reviewer 1 Report

The authors compared the activity and the concentration of carbonic anhydrase VI (CA VI) in biofilm of children with and without caries. This study is logically planned, and the data were carefully statistically analyzed. The overall writing is clear. However, minor revision in the format is required for the following parts.

1.      The names of bacteria should be italic, such as Streptococcus.

2.      In scientific plots, the vertical axes should be titled. The title of the axis should be next to the numbers.

3.      In line 81, “The means and standard deviations of dental caries for the caries group were of 3.162 ± 1.385.” What measurement do the numbers represent? pH?

Author Response

Response to Reviewer 1 Comments

Point 1: The names of bacteria should be italic, such as Streptococcus.

Response 1: The names of bacteria were written in italics along the manuscript.

Point 2: In scientific plots, the vertical axes should be titled. The title of the axis should be next to the numbers.

Response 2: The vertical axes were titled and axes are now close to the numbers.

Point 3: .      In line 81, “The means and standard deviations of dental caries for the caries group were of 3.162 ± 1.385.” What measurement do the numbers represent? pH?

Response 3: The numbers represent the mean and standard deviation of decayed, missing and filled teeth of the group with caries (DMFT index), including the early caries lesion (ECL).  Caries diagnosis was carried out using the visual inspection method with the aid of a clinical mirror, a CPITN probe and flashlight after cleaning and drying the teeth with gauze, according to the World Health Organization criteria, and adding the ECL, as described in sub-section 4.3 of the Materials and Methods section. This sentence was rewritten to: “The mean and standard deviation of the number of decayed, missing and filled teeth for the group with caries were of 3.162 ± 1.385”.

Reviewer 2 Report

 1.  The mat-met section must be completed. It is necessary to understand on which criteria the subjects were classified in the group “caries-free” or in the “caries” group.
2.  Furthermore, it is important to describe in the results section the characteristics of each group according to the criteria of classification of each group.
3.  Please explain, how many caries, one subject of the “caries” group has? Effectively, you explain that “Biofilm samples (about 1 mg) were obtained with a sterile inoculation loop from all accessible buccal surfaces of incisors and molars”. So, if carious lesion are localized on premolars, the biofilm is not associated with carious lesion. Could you clarify this point? Why didn’t you take the biofilm from the teeth associated with carious lesions?
4.  It will be interesting to analyze if the number of caries and the location impact on the CA VI activity and concentration. Others factors could be also studied.
5.  Please could you include as supplementary file, the original data?
6.  Please correct:
a.     Line 35 The name of bacteria “streptococcus mutans” must be write in italics
b.     Line 86 Figure 1 Please add the signification of “*” in the legend

c.     Line 111 Table 1 “biofilme » Please delete the « e » 

d.    Line 159 “r= -0.3910;” Please be consistent, delete the space after the “=”

e.    L228 « be on As a cross-sectional investigation was conducted, “ Please delete “be on”

Author Response

Response to Reviewer 2 Comments

Point 1: The mat-met section must be completed. It is necessary to understand on which criteria the subjects were classified in the group “caries-free” or in the “caries” group. 

Response 1: Caries-free Children - the number of decayed missing or filled teeth (DMFT) was = 0. This means that they had no caries lesion neither early caries lesions(not cavitated), nor cavitated caries lesions according to the World Health Organization criteria and the diagonosis of early caries lesions. On the other side, the Carie group children had a DMFT ≥ 1. This information was added to material and methods heading( Lines 232 – 234).

Point 2: .  Furthermore, it is important to describe in the results section the characteristics of each group according to the criteria of classification of each group.

Response 2: The following description was added in the begining of results heading:

2.1. The DMFT index

The mean and standard deviation of the number of decayed, missing and filled teeth for the group with caries were 3.162 ± 1.385. According to the World Health Organization, this caries index is considered moderate.The caries-free group had no caries lesion.

Point 3: Please explain, how many caries, one subject of the “caries” group has? Effectively, you explain that “Biofilm samples (about 1 mg) were obtained with a sterile inoculation loop from all accessible buccal surfaces of incisors and molars”. So, if carious lesion are localized on premolars, the biofilm is not associated with carious lesion. Could you clarify this point? Why didn’t you take the biofilm from the teeth associated with carious lesions?

Response 3: Biofilm samples were collected from all accessible buccal surfaces of incisors and molars whether they were sound or decayed. Please note that as children were 7 to 9 years old, it is not expected that they present premolars. Regarding the examination of biofilm from early caries lesion and cavitated caries lesion, there is a possibility that activity of CA VI may change/increase according to the caries severity and this information would provide additional insight on how this isoenzyme behaves. To ansewer this question, a longitudinal study is on going in ur laboratory.

Point 4: It will be interesting to analyze if the number of caries and the location impact on the CA VI activity and concentration. Others factors could be also studied.

Response 3: We totally agree with reviewer. However, as shown in table 1 neither CA VI activity nor CA VI concentration showed any correlation with caries. In the same way, CA VI activity in biofilm could to predict caries development.

Point 5: Please could you include as supplementary file, the original data?

Response 3: The original data was included as supplementary file.

Point 6: Please correct:

a.       Line 35 The name of bacteria “streptococcus mutans” must be write in italics

        Response a: This change was performed in the manuscript.

b.      Line 86 Figure 1 Please add the signification of “*” in the legend

Response b: This change was performed in the manuscript.

c.       Line 111 Table 1 “biofilme » Please delete the « e » 

Response c: This change was performed in the manuscript.

d.      Line 159 “r= -0.3910;” Please be consistent, delete the space after the “=”

Response d: The space was deleted.

               e.    L228 « be on As a cross-sectional investigation was conducted, “ Please delete “be on”

                  Response e: This change was performed in the manuscript.

Round 2

Reviewer 2 Report

Please add in the discussion section as a limitation, a part of the Response 3: Biofilm samples were collected from all accessible buccal surfaces of incisors and molars whether they were sound or decayed. Please note that as children were 7 to 9 years old, it is not expected that they present premolars. Regarding the examination of biofilm from early caries lesion and cavitated caries lesion, there is a possibility that activity of CA VI may change/increase according to the caries severity and this information would provide additional insight on how this isoenzyme behaves. To ansewer this question, a longitudinal study is on going in ur laboratory.

Please in the supplementary file, I don't understand why the total number of sample is different between "CA VI activity in biofilm", "CA VI concentration in biofilm" and "Biofilm pH"? Why in "Biofilm pH", the number of "caries free" and "caries" are different?

Author Response

Point 1: Please add in the discussion section as a limitation, a part of the Response 3: Biofilm samples were collected from all accessible buccal surfaces of incisors and molars whether they were sound or decayed. Please note that as children were 7 to 9 years old, it is not expected that they present premolars. Regarding the examination of biofilm from early caries lesion and cavitated caries lesion, there is a possibility that activity of CA VI may change/increase according to the caries severity and this information would provide additional insight on how this isoenzyme behaves. To ansewer this question, a longitudinal study is on going in ur laboratory.

Response 1: The sentence was added in the discussion section as a limitation.

Point 2: Please in the supplementary file, I don't understand why the total number of sample is different between "CA VI activity in biofilm", "CA VI concentration in biofilm" and "Biofilm pH"? Why in "Biofilm pH", the number of "caries free" and "caries" are different?

Response 2: After the biofilm samples were collected, pH, CA VI activity and CA VI concentration were analyzed in this order. As pH analysis was the first, the number of samples was slightly higher because all samples collected were present. In the subsequent analyzes of CA VI activity and concentration, we lost some samples in the pilot studies, in order to calibrate the technique. The concentration analysis was the most difficult in standardization of the technique, so the loss of samples was greater.

In the statistical analysis of pH, the numbers that contained a "*" were more distant from the curve and consequently were excluded from the statistical analysis to reduce the standard deviation, leaving exactly the n = 37 for each group, caries and caries free.